# Serum proteomics reveals disorder of lipoprotein metabolism in sepsis

Xi Liang[1,*], Tianzhou Wu[1,*], Qi Chen[1,*] , Jing Jiang[1,2], Yongpo Jiang[3], Yanyun Ruan[1] , Huaping Zhang[4], Sheng Zhang[3], Chao Zhang[5], Peng Chen[5], Yuhang Lv[4], Jiaojiao Xin[1,2], Dongyan Shi[1,2], Xin Chen[1,6], Jun Li[1,2] , Yinghe Xu[4] 

**Sepsis is defined as an organ dysfunction syndrome and it has high mortality worldwide. This study analysed the proteome of serum from patients with sepsis to characterize the pathological mechanism and pathways involved in sepsis. A total of 59 patients with sepsis were enrolled for quantitative proteomic analysis. Weighted gene co-expression network analysis (WGCNA) was performed to construct a co-expression network specific to sepsis. Key regulatory modules that were detected were highly correlated with sepsis patients and related to multiple functional groups, including plasma lipoprotein particle remodeling, inflammatory response, and wound healing. Complement activation was significantly associated with sepsis-associated encephalopathy. Triglyceride/cholesterol homeostasis was found to be related to sepsis-associated acute kidney injury. Twelve hub proteins were identified, which might be predictive biomarkers of sepsis. External validation of the hub proteins showed their significantly differential expression in sepsis patients. This study identified that plasma lipoprotein processes played a crucial role in sepsis patients, that complement activation contributed to sepsis-associated encephalopathy, and that triglyceride/cholesterol homeostasis was associated with sepsis-associated acute kidney injury.**

## Introduction

Sepsis is a lethal condition defined as an organ dysfunction syndrome caused by an uncontrolled inflammatory response to infection. Sepsis is a leading cause of mortality in the intensive care unit (ICU), with high mortality rates of 10–50%, depending on age and disease severity (Abe et al, 2020; Markwart et al, 2020). Despite the high mortality and burden on the health-care service system, there are few treatments proven to be effective for this syndrome (Rhee et al, 2019). Thus, novel biomarkers with high sensitivity and specificity may be helpful for the diagnosis of sepsis and development of new therapies.

Different tools have been used to investigate the molecular mechanisms of sepsis, including proteomics, genomics, transcriptomics, and metabolomics (Wong et al, 2009; Tang et al, 2010; Siqueira-Batista et al, 2012; Wong, 2012; Skibsted et al, 2013). In particular, liquid chromatography–tandem mass spectrometry (LC–MS/MS)–based proteomics, which focuses on the whole protein complements of organisms, tissues, and cells, could provide high-throughput analysis of proteins, allowing identification or quantification in a single analysis. Several studies have used proteomic analysis to investigate specific biomarkers regulating the pathogenic mechanism of sepsis (Nguyen & Yaffe, 2003; Middleton et al, 2019; Seymour et al, 2019). These studies were performed by differential proteomic analysis followed by enrichment analyses to establish functional pathways and were based on common laboratory indicators or low-throughput techniques. Sepsis is a heterogeneous syndrome, and its pathogenesis involves thousands of proteins. Identification of co-expression patterns might provide in-depth knowledge into sepsis-associated biological pathways.

Weighted gene co-expression network analysis (WGCNA) is a systems biology approach used for finding gene clusters with highly correlated expression levels and for relating them to phenotypic traits (Langfelder & Horvath, 2008). Rather than relating thousands of genes to a clinical trait, WGCNA focuses on the relationship between a few modules and the trait (Zhang & Horvath, 2005; Horvath et al, 2006). Recently, WGCNA has been used to explore co-expression patterns in cancer, ischaemic stroke and sepsis (Dong et al, 2019; Niemira et al, 2019; Wang et al, 2020; Zhang et al, 2020). Thus, in this study, we characterized the proteomic profile in patients with sepsis and identified protein co-expression modules by using WGCNA that could be used in the diagnosis of sepsis.

[1]Precision Medicine Center, Taizhou Central Hospital (Taizhou University Hospital), Taizhou, China   [2]State Key Laboratory for Diagnosis and Treatment of Infectious Diseases, Collaborative National Clinical Research Center for Infectious Diseases, The First Affiliated Hospital, Zhejiang University School of Medicine, Hangzhou, China [3]Department of Intensive Care Unit, Taizhou Hospital of Zhejiang Province, Wenzhou Medical University, Taizhou, China   [4]Department of Intensive Care Unit, Taizhou Central Hospital (Taizhou University Hospital), Taizhou, China   [5]Department of Intensive Care Unit, Taizhou Enze Medical Center (Group) Enze Hospital, Taizhou, China [6]Institute of Pharmaceutical Biotechnology, Zhejiang University School of Medicine, Hangzhou, China

Correspondence: xuyh@tzc.edu.cn; lijun2009@zju.edu.cn; xinchen@zju.edu.cn
*Xi Liang, Tianzhou Wu, and Qi Chen contributed equally to this work

# Results

## Patients and clinical characteristics

A total of 114 patients with sepsis and 62 healthy normal controls (NCs) were enrolled in this study, 90 of whom were placed in the derivation group (59 patients with sepsis and 31 NC), and the remaining subjects were included in the validation group (Fig 1). The clinical characteristics of patients with sepsis at admission and NC subjects who were in the derivation group are summarized in Table S1. Age and sex did not differ between the patients with sepsis and NC ($P > 0.05$). The SOFA score was 6.0 (4.0, 9.0), and the APACHE II score was 18.5 (11.0, 22.0). Among these patients, one patient had a gram-positive bacterial infection, more than 37.5% had a gram-negative bacterial infection, and one patient had a viral infection. The short-term (28/90 d) mortality of the patients with sepsis was 11.9%/11.9%. The levels of laboratory indices, including white blood cell count, haemoglobin level, haematocrit, platelet count, albumin level, aspartate aminotransferase level, and creatinine level, were significantly worse in the sepsis patients compared to the NC group.

## Proteomic characteristics of serum from patients with sepsis

A total of 879 proteins were identified (Table S2). And 396 proteins with 40% or fewer missing values by label-free quantification were used in subsequent analysis. A principal component analysis plot based on the abundance profile of the 396 proteins showed clear separation between sepsis group and NC group (Fig 2A). To determine whether there were distinct patterns of protein expression in patients with sepsis, differential expression analysis of proteomic profiles of sepsis and NC groups was performed using a Benjamini-Hochberg adjusted filter of <0.05 and an expression

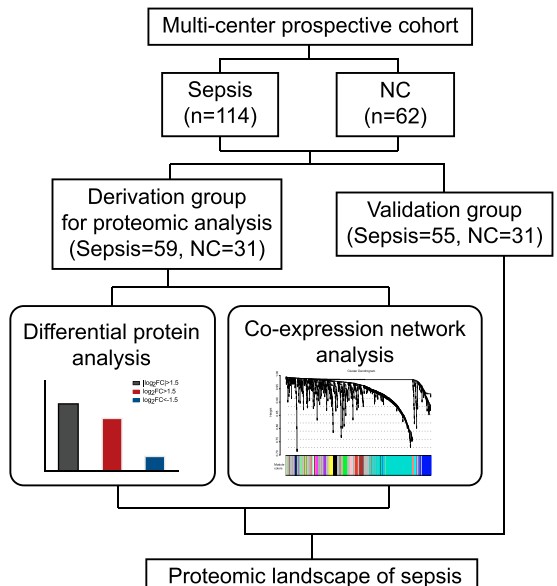

**Figure 1. Overview of the study design and patient group allocation.**
Proteomic analysis was used to identify sepsis-related marker candidates, which were validated on an external group using ELISA.

difference of 1.5-fold or more (Fig 2B). Subsequently, 62 differentially expressed proteins (DEPs) were identified. Of these proteins, 49 (79.0%) were up-regulated, and 13 (21.0%) were down-regulated (Fig S1 and Table S3). As shown in Fig 2C–E, the significantly altered biological process gene ontology (GO) terms acute-phase response, extracellular matrix organization, and negative regulation of endothelial cell apoptotic process were highly enriched with DEPs. The cellular component GO terms extracellular exosome/space/region/matrix were highly enriched with DEPs. The molecular function GO terms cell adhesion molecule binding, antioxidant activity and receptor binding were significantly enriched with DEPs.

## Construction of a sepsis protein co-expression network

A protein co-expression network was constructed using WGCNA based on pairwise protein expression correlations extracted from the matrix of protein expression values. A total of 18 protein high co-expression modules with common expression patterns across the cases analysed were identified, which are shown in different colours (Fig 3A). The size of these modules ranged from 89 proteins (MEgrey) to 5 proteins (MElightcyan). Two methods were implemented to test the relevance between each module and the disease status. Greater module significance was considered to correspond to a greater connection between modules and the disease status, and the result showed that the significance of the MEturquoise module was the highest among the modules (Fig 3B). The correlation between module membership and the disease phenotypes was calculated. The MEturquoise module was highly correlated with disease status, with a correlation coefficient of 0.93 (Fig 3C). Based on these two results, the MEturquoise module was considered the most relevant module to sepsis.

## Functional enrichment analysis of the module most relevant to sepsis

The MEturquoise module was the key module in this study and was most relevant to sepsis. To elucidate the pathogenesis of sepsis, functional enrichment analysis was performed to enrich the biological processes of eigenproteins of MEturquoise. As a result, 202 GO terms were identified and clustered into 32 groups (Fig 4). We found that eigenproteins of MEturquoise were enriched in plasma lipoprotein particle remodeling, regulation of lipid transport, cholesterol biosynthetic processes, acute inflammatory responses, regulation of wound healing, cellular detoxification and regulation of the apoptotic signaling pathway. These results indicated that regulation of plasma lipoprotein particle levels and lipid metabolism might be central to the pathophysiology of sepsis.

## Alteration of the protein network module in sepsis patients with organ dysfunction

To explore the specificity of the network changes for patients with sepsis-associated encephalopathy (SAE), the correlation between module membership and different encephalopathy phenotypes was calculated (Fig 5A [left]). The results showed that the MEyellow module was highly associated with SAE patients. The eigenprotein value of the MEyellow module was significantly changed across

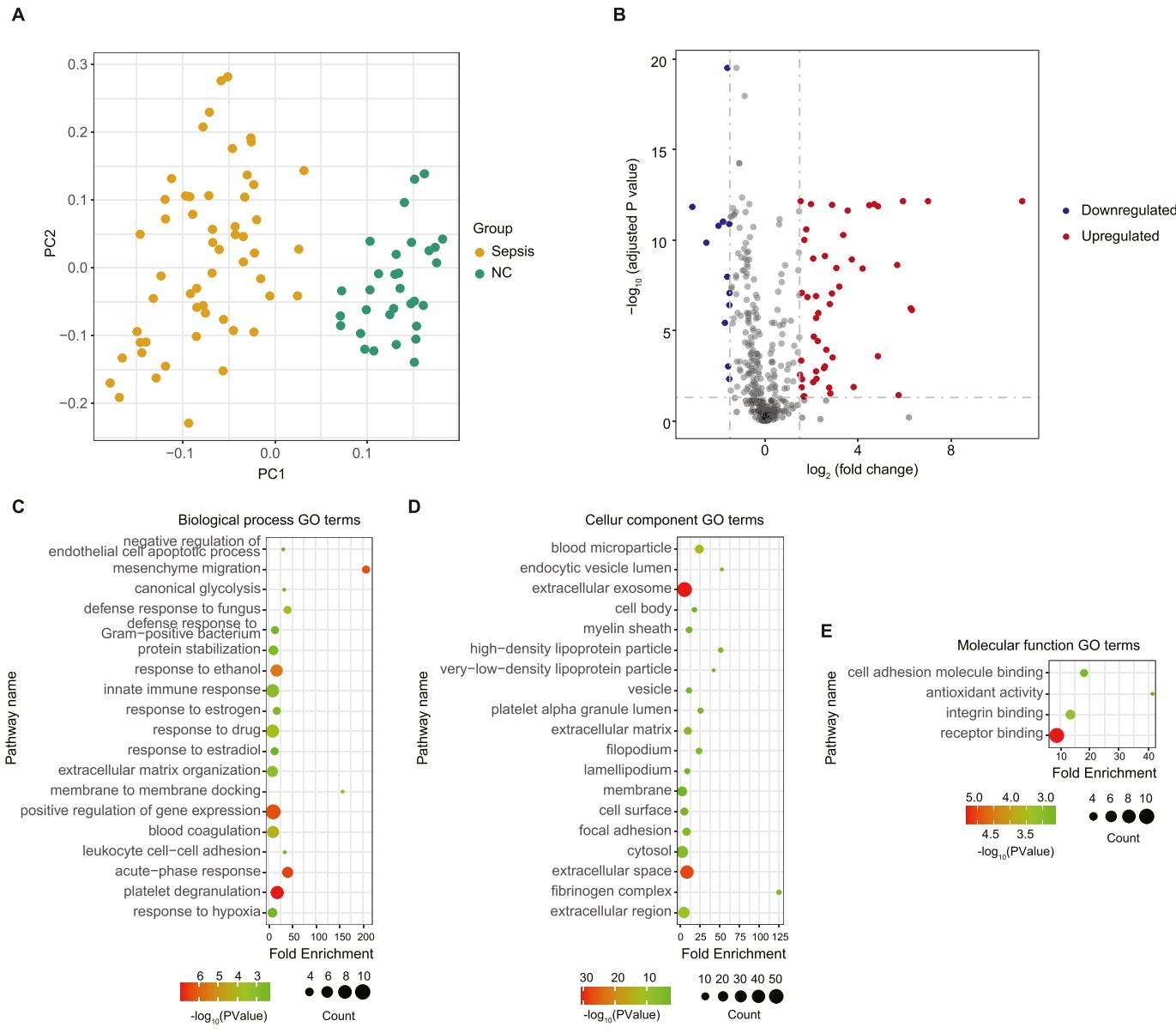

**Figure 2. Proteomic landscape of serum from patients with sepsis and healthy controls.**
**(A)** Principal component analysis showing clear separation between patients with sepsis and healthy controls. **(B)** Volcano plot of differentially expressed proteins, with up- and down-regulated expression coloured red and blue respectively. **(C, D, E)** Summary of functional annotation for sepsis by gene ontology biological processes (C), cellular components (D), and molecular functions (E) related to the differentially expressed proteins.

patients with SAE, patients without SAE and NC (Fig 5A [right upper]). Functional enrichment analysis of the MEyellow module was performed to reveal the pathophysiological changes of patients with SAE. The results illustrated that GO terms related to complement activation were the main enriched biological processes (Fig 5A [right bottom]). The immune response and cell adhesion-related biological processes were also among the proteins enriched in the MEyellow module.

The results of the correlation analysis showed that the MEred module was highly correlated with sepsis-associated acute kidney injury (AKI), in contrast to MEturquoise (Fig 5B [left]). The eigen-protein value of the MEred module was significantly altered across the different phenotypes of AKI (Fig 5B [right upper]). Functional

enrichment analysis showed that lipoprotein metabolic process, triglyceride homeostasis, and negative regulation of endopeptidase activity were enriched with proteins of the MEred module (Fig 5B [right bottom]).

## Identification and validation of sepsis-associated proteins

A total of 81 proteins were connected to the MEturquoise module. A scatterplot showed the correlation of module memberships of these proteins in the MEturquoise module versus the significance of these proteins for sepsis status (correlation coefficient = 0.79, $P = 9.9 \times 10^{-86}$) (Fig 6A). To further investigate the key regulatory

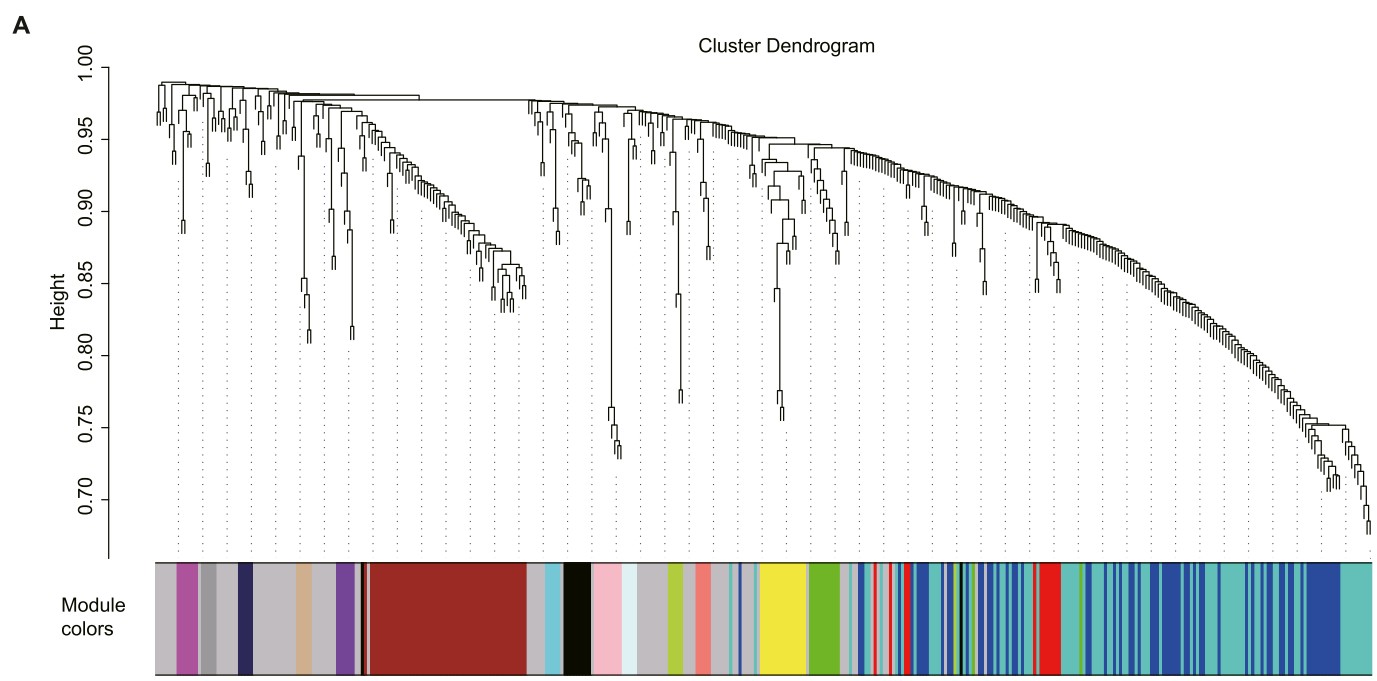

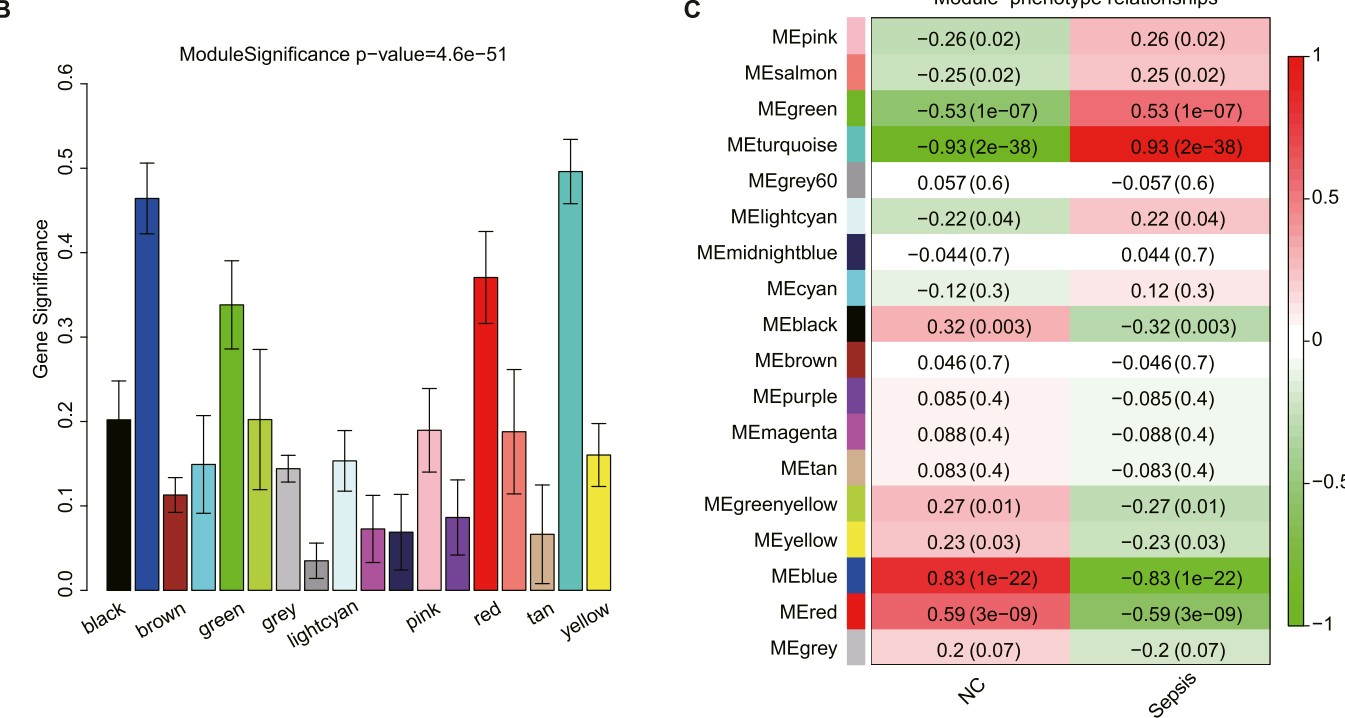

**Figure 3.   Module detection and protein co-expression network construction using weighted gene co-expression network analysis.**
**(A)** Cluster dendrogram and 18 distinct protein co-expression modules defined by dendrogram branch cutting of all proteins of patients with sepsis and healthy controls. **(B)** Bar plot of the significance of each module associated with sepsis. **(C)** Heat map representation of Pearson's correlation between module eigenproteins and different phenotypes.

proteins of the MEturquoise module, a weighted sub-network of proteins in the module was constructed. The results showed that 12 proteins were identified as hub proteins of the network (Fig 6B), and they were specifically expressed in the patients with sepsis (Fig 6C).

To further increase the reliability of the results, four proteins (CRP, LBP, A2GL, and SAA1) were randomly selected for validation using ELISA. The clinical characteristics of the patients with sepsis in the external validation group were similar to those of the

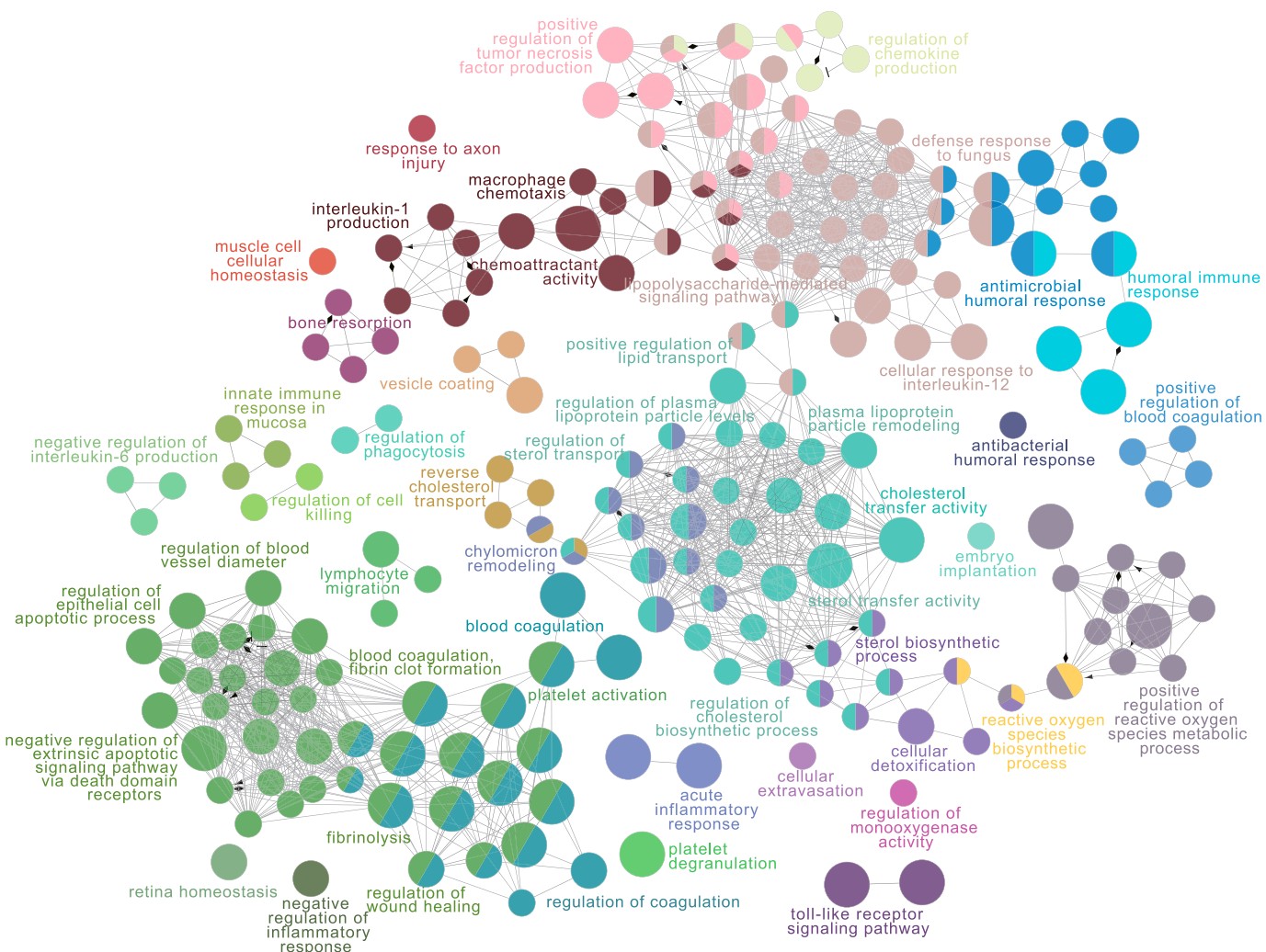

**Figure 4. The pathway network of significantly enriched gene ontology terms in the MEturquoise module.**
Each node represents a gene ontology term. The size of the node represents the significance of the node. Different functional groups are marked with different colours.

patients in the derivation group (Table 1). The results showed that the expression pattern of the four proteins was consistent with that in the initial screening analysis (Fig 6D and E). These observations confirmed that the expression levels of hub proteins are highly specific and can be used as sensitive biomarkers for diagnosing patients with sepsis.

## Discussion

Sepsis is a clinical emergency due to multiorgan dysfunction, which carries a high risk of mortality (Rhodes et al, 2017). Elucidating the pathological processes and identifying the diagnostic biomarkers of sepsis could help to reduce mortality in the clinic. In this study, we analysed 90 serum samples from patients with sepsis and NC subjects by LC–MS/MS-based proteomics. Differential expression analysis and co-expression network analysis were used to view the proteomic changes that occurred during progression from the

normal to sepsis state. According to ELISA-based validation, our proteomic analysis results had high reliability and quality.

Based on differential expression analysis, 62 DEPs were identified between patients with sepsis and NC subjects. Functional enrichment analysis of these DEPs showed that the biological process GO terms inflammatory response, extracellular matrix-related processes, and reactive oxygen species-related processes were significantly enriched. The extracellular matrix plays an important role in the migration of leukocytes from the bloodstream to sites of inflammation (Lorente et al, 2009). Reactive oxygen species induce renal tubular injury or renal vascular injury during sepsis (Schrier & Wang, 2004). The results aligned with the characteristic pathogenesis of sepsis.

In this study, WGCNA was used to construct a protein co-expression network. WGCNA is a useful algorithm that can identify similar expression pattern modules and investigate the complex relationship between these modules and clinical traits (Barabási et al, 2011). Pathway changes across the different disease status can

**A**

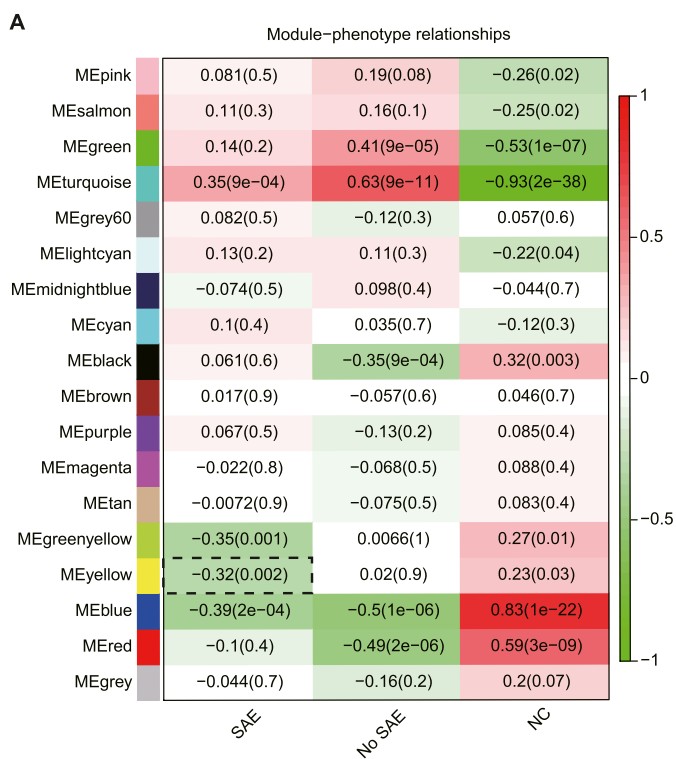

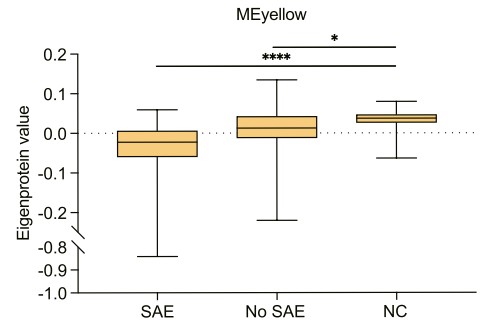

| Term | PValue | Protein count |
|------|--------|---------------|
| GO:0030449~regulation of complement activation | 0 | 7 |
| GO:0006956~complement activation | 0 | 8 |
| GO:0019835~cytolysis | 0 | 6 |
| GO:0006958~complement activation, classical pathway | 0 | 7 |
| GO:0006957~complement activation, alternative pathway | 0 | 5 |
| GO:0001867~complement activation, lectin pathway | 0 | 3 |
| GO:0010951~negative regulation of endopeptidase activity | 0 | 4 |
| GO:0006955~immune response | 0.008 | 3 |
| GO:0010951~negative regulation of endopeptidase activity | 0.015 | 4 |

**B**

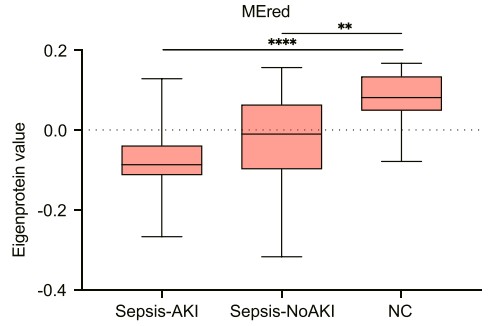

| Term | PValue | Protein count |
|------|--------|---------------|
| GO:0042157~lipoprotein metabolic process | 0 | 5 |
| GO:0070328~triglyceride homeostasis | 0 | 4 |
| GO:0010951~negative regulation of endopeptidase activity | 0 | 5 |
| GO:0034382~chylomicron remnant clearance | 0 | 3 |
| GO:0034372~very-low-density lipoprotein particle remodeling | 0 | 3 |
| GO:0001523~retinoid metabolic process | 0 | 4 |
| GO:0042632~cholesterol homeostasis | 0 | 4 |
| GO:0033700~phospholipid efflux | 0 | 3 |
| GO:0043691~reverse cholesterol transport | 0.001 | 3 |
| GO:0019433~triglyceride catabolic process | 0.006 | 3 |
| GO:0033344~cholesterol efflux | 0.008 | 3 |

**Figure 5. Protein network module changes in patients with sepsis-associated encephalopathy (SAE) and sepsis-associated acute kidney injury (AKI).**
**(A)** (Left) Heat map representation of the relationship between module eigenproteins and different phenotypes of SAE. (Right upper) Synthetic eigenprotein analysis for the pink module, which is highly correlated with SAE patients, except the turquoise module. (Right bottom) Gene Ontology terms enriched in pink modules. **(A, B)** Similar to (A), association of network modules in sepsis-associated AKI. (Left) Heat map representation of the relationship between module eigenproteins and different phenotypes of AKI. (Right upper) Synthetic eigenprotein analysis for the black module, which is highly correlated with sepsis-associated AKI patients, except the turquoise module. (Right bottom) Gene ontology terms enriched in the black module. Data information: In (A, B), data are presented as median with IQR. ****$P$-value < 0.0001; ***$P$-value < 0.001; **$P$-value < 0.01; *$P$-value < 0.05 (Kruskal–Wallis test).

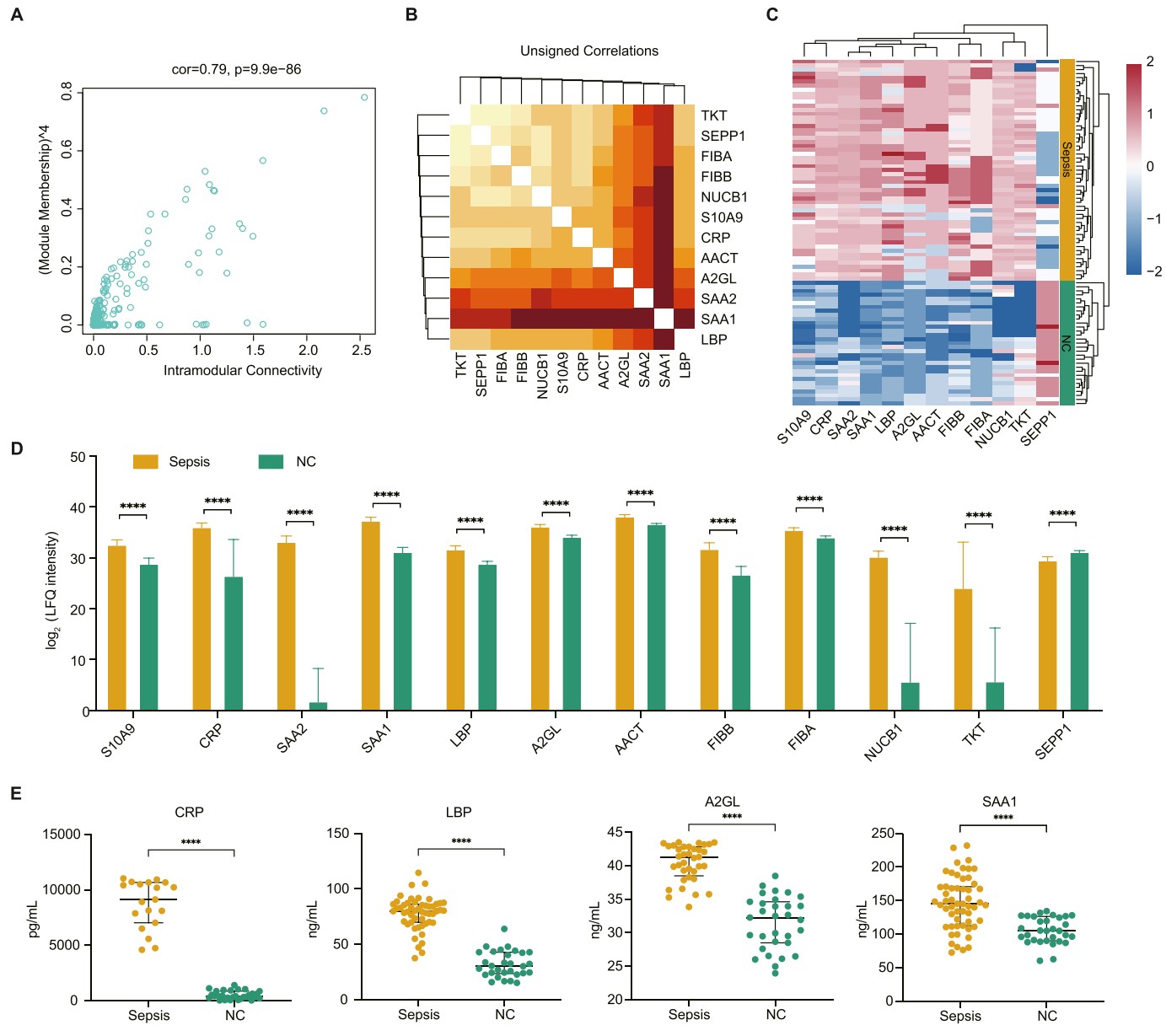

**Figure 6. Sepsis-related potential biomarkers from the module highly correlated to sepsis.**
**(A)** Scatter plot of module eigenproteins in the turquoise module. **(B)** Heat map representation depicting the topological overlap matrix among the 12 hub proteins. Light colours represent low overlap, and progressively darker red represents higher overlap. **(C)** Unsupervised hierarchical cluster analysis of the abundance profile of the 12 hub proteins in patients with sepsis and healthy controls. Label-free quantification protein intensities were $log_2$-transformed and normalized into Z-scores by rows. **(D)** The abundance of the 12 hub proteins in the derivation group. **(E)** ELISA-based validation of four randomly selected biomarkers of sepsis development. **(D, E)** Data information: In (D, E), data are presented as the mean ± SD or median with IQR ****$P$-value < 0.0001; ***$P$-value < 0.001; **$P$-value < 0.01; *$P$-value < 0.05 (Mann–Whitney U test).

be recognized, providing new insight into biological networks related to clinical traits of interest. Proteins in the same module are considered to be functionally associated with each other. The other notable advantage of WGCNA is that it can identify the interactions among proteins in different co-expression modules. WGCNA has been used to explore co-expression patterns in different fields, such as cancer (Thorsson et al, 2018) and Alzheimer's disease (Johnson et al, 2020), and is useful in identifying potential signature clusters or biomarkers of targeted phenotypic traits (Chou et al, 2014).

The turquoise module was found to be strongly correlated with patients with sepsis by WGCNA. Notably, processes related to plasma lipoprotein and lipid metabolism were the most changed functional categories of the turquoise module, illustrating that lipoprotein and lipid metabolism might play a significant role in patients with sepsis. Lipoprotein particles protect the endothelium (Rubin et al, 1991), and they could play crucial antiapoptosis and antioxidation roles. In addition, lipoproteins are reportedly involved in innate immunity (Remmerie & Scott, 2018). Changes in lipoproteins are also related to a variety of inflammatory disorders

**Table 1. Characteristics of enrolled patients with sepsis included in the derivation group and validation group.**

| | Derivation group | Validation group | *P*-value |
|---|---|---|---|
| N | 59 | 55 | |
| Male (%) | 33 (55.9%) | 31 (56.4%) | 0.696 |
| Age (yr) | 71.0 (61.0, 78.0) | 75.0 (64.0, 84.7) | 0.067 |
| Laboratory data | | | |
| Mean arterial pressure (mm Hg) | 79.3 (71.5, 90.2) | 77.6 (68.7, 90.0) | 0.610 |
| White blood cell count ($10^9$/L) | 12.8 (7.6, 18.4) | 12.4 (7.4, 18.4) | 0.933 |
| Haemoglobin (g/l) | 114.5 (92.7, 127.0) | 107.0 (83.0, 129.5) | 0.432 |
| Haematocrit (%) | 33.9 (28.3, 38.3) | 32.3 (26.3, 39.3) | 0.570 |
| Platelet count ($10^9$/L) | 136.0 (68.3, 198.7) | 133.0 (73.5, 174.0) | 0.917 |
| Albumin (g/dl) | 27.2 (25.0, 29.95) | 27.0 (24.4, 30.8) | 0.041 |
| Aspartate aminotransferase (U/l) | 46.0 (26.5, 127.5) | 40.0 (24.0, 82.0) | 0.614 |
| Alanine aminotransferase (U/l) | 30.0 (18.0, 68.0) | 21.0 (14.0, 42.0) | 0.144 |
| Total bilirubin ($\mu$mol/l) | 15.8 (8.0, 26.1) | 15.9 (10.9, 44.2) | 0.414 |
| Creatinine ($\mu$mol/l) | 135.0 (95.0, 218.0) | 103.0 (73.5, 199.5) | 0.078 |
| INR | 1.2 (1.1, 1.3) | 1.3 (1.1, 1.4) | 0.818 |
| Infection | | | 0.832 |
| Gram-positive bacteria (%) | 7 (11.9%) | 3 (5.5%) | |
| Gram-negative bacteria (%) | 24 (40.7%) | 23 (41.8%) | |
| Viral (%) | 1 (1.7%) | 1 (1.8%) | |
| Other (%) | 27 (45.7%) | 28 (50.9%) | |
| CRRT | 8 (13.6%) | 7 (12.7%) | 0.246 |
| Vasopressors | | | 0.250 |
| No (%) | 32 (54.2%) | 33 (60.0%) | |
| Yes (%) | 25 (42.4%) | 22 (40.0%) | |
| Unknown (%) | 2 (3.4%) | 0 (0.0%) | |
| Mechanical ventilation | | | 0.555 |
| No (%) | 34 (57.6%) | 29 (52.7%) | |
| Yes (%) | 22 (37.3%) | 25 (45.5%) | |
| Unknown (%) | 3 (5.1%) | 1 (1.8%) | |
| Severity at time of intensive care unit admission | | | |
| SOFA | 6.0 (4.0, 8.7) | 6.0 (4.0, 8.0) | 0.972 |
| APACHE II | 18.0 (11.0, 22.0) | 18.0 (13.0, 23.0) | 0.293 |
| Mortality | | | |
| 28-d | 8 (13.6%) | 10 (18.2%) | 0.7 |
| 90-d | 8 (13.6%) | 11 (20.0%) | 0.39 |

Data are expressed as the mean ± SD, median (IQR) or number of patients (percentages). Continuous variables were compared by using *t* test and the Mann-Whitney U test, and categorical variables were compared by using the χ2 or Fisher's exact test between the discovery and validation groups. SOFA, Sequential Organ Failure Assessment on day of sampling. APACHE II, Acute Physiology and Chronic Health Evaluation II. CRRT, Continuous Renal Replacement Therapy.

(van Leeuwen et al, 2003). A recent study revealed that oxidized low-density lipoprotein along with lipopolysaccharide could induce monocytes to promote inflammasome-mediated adaptive immunity (Christ et al, 2018). Overall, lipoprotein and lipid metabolism might be central to the pathophysiology of patients with sepsis. In addition, the inflammatory response, wound healing, blood coagulation and the apoptotic signaling pathway were also identified.

These results suggested that co-expression network analysis could identify a set of proteins with similarly altered expression not restricted by the significance of the changes.

The expression level of the yellow module increased in the patients with SAE. The yellow module was mainly enriched in complement activation, suggesting that complement-related immunity may drive the main pathophysiology of patients with SAE. This finding

is consistent with previous research, which reported that complement system activation contributes to blood–brain barrier function in experimental sepsis (Flierl et al, 2009). High-density lipoprotein cholesterol has been linked to the risk of sepsis-associated AKI (Roveran Genga et al, 2017). Secreted zinc-dependent endopeptidases were correlated to the severity of kidney insufficiency (Zuo et al, 2021). Further information on patients with SAE and sepsis-associated AKI should be clarified in future studies.

The hub proteins within co-expression modules most associated with disease biology might be key drivers of disease pathogenesis (Huan et al, 2013, 2015). A total of 12 hub proteins were identified as candidates within the turquoise module, which were tightly associated with the prognosis of sepsis. CRP has been widely used in clinical practice for sepsis, and it is associated with an increased risk of organ failure and death (Pierrakos & Vincent, 2010). Studies have reported that SAA1 and SAA2 are involved in the biological process of lipoprotein/cholesterol metabolism during the acute-phase response (Annema et al, 2010). And SAA is correlated with CRP in patients with septic shock (Orro et al, 2004; Cicarelli et al, 2008). LBP is involved in the proinflammatory response to gram-negative bacterial infections (Stasi et al, 2017). SAA and LBP serve as traditional biomarkers for sepsis (Mussap et al, 2013). NUCB1 plays an important function in the transport of matrix metalloproteinases (Pacheco-Fernandez et al, 2020), which are regulators of inflammatory processes. Fibrinogen plays a critical role in the coagulation cascade and inflammation (Davalos & Akassoglou, 2012; Feng et al, 2013). Fibrinogen alpha chain FIBA and fibrinogen beta chain FIBB encode the $\alpha$ and $\beta$ subunits of the fibrinogen. A previous study has reported FIBB haplotype association with mortality and organ dysfunction in sepsis (Manocha et al, 2007). Interestingly, a lack of SEPP1 induced neurological dysfunction in the mice, and SEPP1 can deliver Se to neurons (Solovyev et al, 2018), illustrating that it might have importance in patients with SAE. S10A9 is plentiful cytoplasmic proteins of phagocytes, which has been reported to amplify phagocyte activation during sepsis (Vogl et al, 2007). A2GL had specific expression in sepsis, which could be as a new biomarker for sepsis (Gong et al, 2020; Lu et al, 2020). AACT as an inflammation-related marker (Saetre et al, 2007; Fillman et al, 2013), has been found to significantly over-expressed expression in sepsis mediated by glucocorticoids (Gueugneau et al, 2018). These hub proteins may contribute to improvement of diagnosis and therapeutic decision-making for patients with sepsis. Our external validation with four randomized proteins confirmed the same expression trends of proteomic analysis.

In summary, based on a proteomic co-expression network, our study revealed that lipoprotein-related pathways were the major altered biological processes in sepsis. The central role of the complement system in patients with SAE was confirmed. The hub proteins highly correlated with biological processes were identified, providing a useful tool for diagnosing sepsis and new insights for understanding the pathophysiology of sepsis.

# Materials and Methods

## Study design

Patients with sepsis were prospectively enrolled from three participating ICUs between 1 April 2019 and 16 August 2020. The last follow-up was completed on 20 November 2020. Adult patients (aged > 18 yr) who were admitted to the ICU with sepsis were enrolled in the study. The blood samples from patients who diagnosed with sepsis were collected at admission of ICU. Detailed clinical and follow-up data for all enrolled patients were collected from the electronic data capture system and case report forms. Sepsis was defined as the presence of an infection combined with an acute change in SOFA score of 2 or more points (Singer et al, 2016). NC subjects (with a SOFA score of 0 and without infection) were recruited as the control group from the Physical Examination Center during the same time. The patients and controls were randomly allocated into a derivation group and a validation group (Fig 1). Differential expression analysis and co-expression network analysis were performed to identify variations in protein levels. Potential biomarkers were validated by ELISA in serum from 55 subjects with sepsis. The study protocol was approved by the Clinical Research Ethics Committee of Taizhou Central Hospital (Taizhou University Hospital) (registration number: 2019-016, principal investigator: Yinghe Xu, date of registration: 26 February 2019). Written informed consent was obtained from all participants or their legal representative.

## Sample preparation for proteomic analysis and ELISA

Blood samples of patients and NC subjects were collected and allowed to clot at room temperature for 60 min. Serum was separated by centrifugation at 1,600$g$ for 10 min within 30 min to remove insoluble solids and stored at –80°C until proteomic analysis and ELISA (Tammen, 2008). Removal of high-abundance proteins in serum, such as albumin and IgG, was performed using ProteoPrep Blue Albumin & IgG Depletion Kit (PROTBA; Sigma-Aldrich) according to the manufacturer's instructions. Removal of impurities from the protein extraction was performed using a 2-D clean kit (GE Healthcare) before the determination of the sample concentration.

## LC−MS/MS analysis

The proteins in serum were digested with sequencing-grade trypsin. The resulting peptide mixtures were subjected to an SMA1000 ultramicro UV spectrophotometer to measure the supernatant concentration. A Waters UPLC system was used to separate peptides with a BEH C18 nano-ACQUITY column (75 $\mu$m × 25 cm, 1.7 $\mu$m). Proteomic analysis was performed using nanoflow liquid chromatography (ACQUITY UPLC system; Waters Co.) coupled with mass spectrometry (Q Exactive mass spectrometer; Thermo Fisher Scientific). The profile mode of an Orbitrap at a resolution of 70,000 was used to acquire a full-mass scan (300−−140 m/z). Peptide segments with charges ranging from +2 to +6 were chosen for further LC−MS/MS analysis. Fragmentation was performed based on high-energy collisional dissociation at 27% normalized collision energy and with a resolution of 35,000, and the 20 most intense peaks were selected. An exclusion duration of 20 s was used for dynamic exclusion.

## Protein identification and database searching

The spectra generated by LC−MS/MS were searched against *Homo sapiens* proteins in the SwissProt database using the MaxQuant

search engine (version 1.6.1.0). The precursor mass was set to 20 ppm, and the fragment ion mass tolerance was set to 20 ppm. Methionine oxidation and protein N-terminal acetylation were set as variable modifications, and carbamidomethyl was set as a fixed modification. Two missed cleavages for trypsin were allowed. A false discovery rate of 1% was set at both the peptide and protein levels to filter the results. MaxQuant with label-free quantification was used for protein quantification. A minimum peptide ratio count of two was required, and only unmodified peptides were used for relative quantification. A total of 879 proteins were identified from serum samples. The expression matrix was normalized by normalizeBetweenArrays function of the R package limma, which is an optimal normalization method commonly used in proteomics data analysis (Zhang et al, 2014). Normalized protein abundance was $\log_2$-transformed and used in all quantitative analysis. The missing values were imputed with the minimum of the proteomic data.

### Protein co-expression network analysis

Network analysis was performed to identify modules of co-expressed proteins. The WGCNA package in R was used to normalize protein abundance to define protein co-expression networks. The function WGCNA:blockwiseModules() was used with the following settings: soft threshold power $\beta$ = 2, deepSplit = 4, minModuleSize = 10, mergeCutHeight = 0.05, threshPercent = 50, and mergePercent = 25, and all other parameters were set to the default. Soft threshold power $\beta$ defined strong correlations between proteins and penalized weak correlations. Module eigenprotein correlation value kME is defined as a module membership measure, which was calculated by Pearson correlations between each protein and each module eigenprotein. The percentage of module members checked for kME overlap of 50% (threshPercent = 50) and the threshold for merging modules with a high common kME.intramodule of 25% (mergePercent = 25) were used to reduce the number of modules. The topological overlap matrix measured the network connectivity of a protein, which is defined as the sum of its adjacency with all other proteins for network generation. Hierarchical clustering analysis was conducted based on 1-topological overlap matrix, with a merge cutoff height of 0.05, to classify proteins with similar expression patterns.

### Identification of clinically significant modules and candidate proteins

Module eigenproteins were used as the first principal component of modules in a given protein expression dataset. To identify modules highly associated with clinical information, Pearson correlation was performed between module eigenproteins and clinical traits. In addition, gene significance was calculated by the $\log_{10}$ transformation of the P-value in the linear regression between each gene expression and clinical information value. Module significance was calculated by the average gene significance of all proteins in a module. The module with first-ranked module significance was considered the one most related to the clinical information. Hub proteins of the clinically significant modules were considered candidate proteins that may play an

important role in the pathophysiology of the disease. Hub proteins in this study were defined by a membership value >0.8 and correlation with clinical traits >0.2.

### Functional enrichment analysis

Functional enrichment analysis was performed on altered proteins and co-expressed modules highly associated with disease status by using DAVID 6.8 (Dennis et al, 2003) and ClueGO (Bindea et al, 2009) to glean a deeper biological understanding of these co-expressed modules. The significance of functional enrichment analysis was defined with a false discovery rate <0.05.

### Validation by ELISA

To further confirm the results of the proteomic analysis, hub proteins of modules that were significantly correlated with clinical traits were randomly validated by ELISAs using external samples in accordance with the manufacturer's instructions. The information of ELISA kits used in this study was provided in Table S4.

### Statistical analysis

The results of the measurements 1 are presented as the mean ± standard deviation (SD) or median with interquartile range (IQR), unless otherwise noted. No statistical method was used to pre-determine sample sizes. Comparisons between patients with sepsis and subjects of NC were performed using Welch's $t$ test to ensure sensitivity, assuming two-tailed distributions. The normality assumption of Welch's $t$ test was validated using the Shapiro–Wilk test when the P-value >0.05 in each compared group. The non-parametric Mann–Whitney U test was used for the non-normally distributed data. The Benjamini–Hochberg procedure was used to correct the false discovery rate to control type I error in multiple tests. Proteins quantified in at least fewer than 40% missing value were included in further analysis. The protein abundance was normalized by normalizeBetweenArrays function of the R limma. $\log_2$ transformation of the normalized protein abundance was performed. The criterion "$\log_2$FC > 1.5, adjusted P-value < 0.05" was used to detect significant changes in protein levels. Comparisons of categorical variables were performed using the $\chi^2$ test.

All statistical analyses were performed using the R software package. Principal component analysis was performed by using the "sva" function. The hub proteins validated via ELISA were identified using the Mann-Whitney U test. The P-value < 0.05 was considered significant.

### Ethics approval and consent to participate

Ethics approval was granted by the Clinical Research Ethics Committee of Taizhou Central Hospital (Taizhou University Medical School) (registration number: 2019-016, principal investigator: Yinghe Xu, date of registration: 26 February 2019).

# Data Availability

The mass spectrometry proteomics data have been deposited to the ProteomeXchange Consortium (http://proteomecentral.proteomexchange.org) via the iProX partner repository with the dataset identifier IPX0003225000 and PXD027485.

# Supplementary Information

# Acknowledgements

This work was supported by the National Natural Science Foundation of China (81830073) and the Science and Technology Project of Taizhou (1801KY70, 1902KY02).

## Author Contributions

X Liang: software, formal analysis, visualization, methodology, and writing—original draft.

T Wu: resources, data curation, and writing—original draft.

Q Chen: resources, data curation, and validation.

J Jiang: data curation, software, formal analysis, and methodology.

Y Jiang: resources, data curation, and funding acquisition.

Y Ruan: software and validation.

H Zhang: resources.

S Zhang: resources.

C Zhang: resources.

P Chen: resources.

Y Lv: resources.

J Xin: resources and validation.

D Shi: resources and validation.

X Chen: conceptualization, data curation, and writing—review and editing.

J Li: data curation, funding acquisition, and writing—review and editing.

Y Xu: resources, funding acquisition, project administration, and writing—review and editing.

## Conflict of Interest Statement

The authors declare that they have no conflict of interest.

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
