## [Reviewer comments · Life Science Alliance]

Life Science Alliance

Serum proteomics reveals disorder of lipoprotein metabolism in sepsis

Xi Liang, Tianzhou Wu, Qi Chen, Jing Jiang, Yongpo Jiang, Yanyun Ruan, Huaping Zhang, Sheng Zhang, Chao Zhang, Peng Chen, Yuhang Lv, Jiaojiao Xin, Dongyan Shi, Xin Chen, Jun Li, and Yinghe Xu

DOI: <https://doi.org/10.26508/lsa.202101091>

Corresponding author(s): Yinghe Xu, Taizhou Central Hospital (Taizhou University Hospital) and Jun Li, The First Affiliated Hospital, Zhejiang University School of Medicine

Review Timeline:

Submission Date:	2021-04-06
Editorial Decision:	2021-06-10
Revision Received:	2021-07-27
Editorial Decision:	2021-08-03
Revision Received:	2021-08-09
Accepted:	2021-08-10

Transaction Report:

June 10, 2021

Re: Life Science Alliance manuscript #LSA-2021-01091

Prof. Yinghe Xu
Taizhou Central Hospital (Taizhou University Hospital)
999 Donghai Avenue
Taizhou 318000

Dear Dr. Xu,

Thank you for submitting your manuscript entitled "Serum proteomics reveals disorder of lipoprotein metabolism in sepsis" to Life Science Alliance. The manuscript was assessed by expert reviewers, whose comments are appended to this letter. We invite you to submit a revised manuscript addressing the Reviewer comments.

Thank you for this interesting contribution to Life Science Alliance. We are looking forward to receiving your revised manuscript.

Sincerely,

- A letter addressing the reviewers' comments point by point.
- An editable version of the final text (.DOC or .DOCX) is needed for copyediting (no PDFs).
- High-resolution figure, supplementary figure and video files uploaded as individual files: See our detailed guidelines for preparing your production-ready images, <https://www.life-science-alliance.org/authors>
- Summary blurb (enter in submission system): A short text summarizing in a single sentence the study (max. 200 characters including spaces). This text is used in conjunction with the titles of papers, hence should be informative and complementary to the title and running title. It should describe the context and significance of the findings for a general readership; it should be written in the present tense and refer to the work in the third person. Author names should not be mentioned.

B. MANUSCRIPT ORGANIZATION AND FORMATTING:

Reviewer #1 (Comments to the Authors (Required)):

The authors performed the weighted gene co-expression network analysis to construct a co-expression network for septic patients. They found that the plasma lipoprotein processes played a crucial role in sepsis patients, the complement activation contributed to sepsis-associated encephalopathy and the metal ion response was associated with sepsis-associated acute kidney injury. Using the comprehensive analyses, the results look solid. In addition, the study is novelty with some clinical significance in sepsis diagnosis.

Comments:

1. How was the sample size determined? Please add statistical analyses details of how the sample size was determined.
2. The manuscript should report the sampling time (e.g. at ICU admission, the time when the sepsis was diagnosed, or some other time points else).
3. In the Table 1, we think it would be better to use "yes" or "no" rather than "1" or "0" for the use of vasopressors and mechanical ventilation.

Reviewer #2 (Comments to the Authors (Required)):

In this manuscript, the authors describe the proteome of serum from 59 patients with sepsis and 31 healthy controls. Bioinformatics analysis finds lipoprotein and lipid metabolism might play a significant role in patients with sepsis. The authors also find some signal pathways associated with SAE and AKI. Based on the bioinformatic analysis, four sepsis-associated proteins which may act as biomarkers of patients with sepsis are validated in another 55 patients with sepsis and 31 healthy controls. The conclusions are mostly supported by the data, a few need to be more discussed. The authors could either add experimental data or carefully re-phrase the respective conclusions.

1. In figure 6B-E, "LRG1", "SAA1" and "SAA2" are analyzed. But in figure 6F, the authors validate "LRG" and "SAA" proteins in serum. "LRG" and "LRG1" are gene names of A2GL protein. The authors did proteomics analysis. Therefore they should ensure that no gene names were included.
2. There are three SAA proteins found in humans: SAA1, SAA2, and SAA4. SAA4 hasn't been identified as a sepsis-associated protein in earlier bioinformatics analysis. The authors should clarify why they test SAA rather than SAA1 or SAA2 although SAA2 seems to be a better biomarker candidate.
3. In figure 5B, "GO: 0006334~nucleosome assembly" is enriched in the top with 14 counts, which is more than the 9 counts of all associated "metal ion response", yet it is not discussed in the manuscript.
4. Please provide the ELISA kit information and sample preparation in the Materials and Methods section.
5. Please supply the explanation of "NC".
6. In Figure 2C, it is suggested that the protein number and meaning of the grey are added as the red and the blue.
7. Please check the typos and grammar mistakes such as: "remodelling", "signalling" etc.

Reviewer #3 (Comments to the Authors (Required)):

The authors describe a serum proteomics analysis of sepsis patients compared to controls. This could be an interesting study but the current version of the manuscript has serious flaws in presenting the proteomics data, and it is not possible to evaluate the quality of the data.

In Results, the proteomics-sections starts from PCA-plot of the results, and the LC-MS/MS results are not mentioned/shown. This is a major problem for the manuscript. These results should be presented as tables/supplementary tables and the results opened up. Further, the raw data should be deposited into a publicly available proteomics repository like PRIDE (ProteomeXchange).

Fig 2B is not very informative in its current form, and Fig 2C is not needed, it just shows the number

of DEPs already mentioned in the results. It would be much more informative to provide details on the DEPs as a table. Fig 3-6 are somewhat difficult to follow and the text in Results related to these is partly repetitive and could be significantly shortened.

Re protein identification and quantification results: how many proteins were identified from serum samples? How reproducible was the depletion method used? In Mat+Met why was precursor mass tolerance 20 ppm, was the MS not properly calibrated? How was the data processed after MaxQuant search? Something is mentioned in 'Statistical analysis' but many details are missing. How e.g were missing values taken into account in the analysis? It is written that if protein was quantified in two samples it was taken into statistical analysis. Thinking on the number of samples in proteomics analysis (90 in total) having a valid value in 2/90 samples is really low posing questions on the quality of the results.

Why did the authors choose proteins for ELISA validation 'randomly'? Is CRP really a random choice and a new finding related to sepsis? How many of the proteins found to be DEPs in this study have previously been linked to sepsis? What is the main novelty of the current study?

Dear Reviewers,

Thank you very much for spending your valuable time reviewing our manuscript. We have tried our best to address the issues raised by each of you and hope that we have addressed all your concerns regarding this work.

Sincerely,

Dr. Yinghe Xu

Response to the comments raised by Reviewer #1

Reviewer #1

Comments

Q1: How was the sample size determined? Please add statistical analyses details of how the sample size was determined.

Response: This study aims to profile the pathological mechanism and pathways involved in sepsis based on serum proteomic in a prospective cohort. No specific indicators or interventions were used to predetermine the sample size. This information has been described in the Method section (Page 20, lines 20-21).

Q2: The manuscript should report the sampling time (e.g. at ICU admission, the time when the sepsis was diagnosed, or some other time points else).

Response: The blood samples from patients who diagnosed as sepsis were collected at admission of ICU. As suggested, the information has been added in the revised manuscript (Page 16, lines 7-8).

Q3: In the Table 1, we think it would be better to use "yes" or "no" rather than "1" or "0" for the use of vasopressors and mechanical ventilation.

Response: As suggested, we have used “yes” or “no” for the use of vasopressors and mechanical ventilation in the revised Table 1 and Supplementary Table 1.

Response to the comments raised by Reviewer #2

Reviewer #2

Comments

Q1: In figure 6B-E, "LRG1", "SAA1" and "SAA2" are analyzed. But in figure 6F, the authors validate "LRG" and "SAA" proteins in serum. "LRG" and "LRG1" are gene names of A2GL protein. The authors did proteomics analysis. Therefore they should ensure that no gene names were included.

Response: Thank you very much for your suggestion. The protein names have been unified, and no gene names were included in the revised manuscript (Page 10, lines 7-8) and Figure 6.

Q2: There are three SAA proteins found in humans: SAA1, SAA2, and SAA4. SAA4 hasn't been identified as a sepsis-associated protein in earlier bioinformatics analysis. The authors should clarify why they test SAA rather than SAA1 or SAA2 although SAA2 seems to be a better biomarker candidate.

Response: Thank you very much for your suggestion. The SAA in the ELISA validation is SAA1 protein, which has been corrected in the revised manuscript (Page 10, line 8) and Figure 6.

The 12 proteins identified in Figure 6B-D were all biomarker candidates. The aim of the validation with ELISA was to confirm the expression value of proteomic analysis. In consideration of the availability of samples and ELISA kits, we randomly selected 4 proteins to validate the reliability of proteomic analysis. This information has been added in the revised manuscript (Page 15, lines 3-4).

Q3: In figure 5B, "GO: 0006334~nucleosome assembly" is enriched in the top with 14 counts, which is more than the 9 counts of all associated "metal ion response", yet it is not discussed in the manuscript.

Response: As another reviewer suggested, we have reanalyzed the proteomic data. The Gene Ontology terms significantly enriched in the sepsis-associated acute kidney injury have been updated in the Figure 5B. And all significantly enriched terms have been discussed in the revised manuscript (Page 13, lines 14-17).

Q4: Please provide the ELISA kit information and sample preparation in the Materials and Methods section.

Response: As you suggested, the information of ELISA kit and sample preparation has been added in the revised Materials and Methods section (Pages 16, line 23; Page 17, lines 3-4; Page 20 lines 15-16).

Q5: Please supply the explanation of "NC".

Response: Thank you for your suggestion. NC is the abbreviation of the healthy normal control group. This information has been added in the revised manuscript (Page 6, line 4).

Q6: In Figure 2C, it is suggested that the protein number and meaning of the grey are added as the red and the blue.

Response: As another reviewer suggested, Figure 2C has been placed in the Supplementary Material as Figure S1 in the revised manuscript and the protein number and meaning of the grey has been added.

Q7: Please check the typos and grammar mistakes such as: "remodelling", "signalling" etc.

Response: Thank you for pointing out our typo, we have corrected these mistakes in the revised manuscript (Page 3, line 10; Page 8, line 13 and 15; Page 13, line 4).

Response to the comments raised by Reviewer #3

Reviewer #3

Comments

Q1: In Results, the proteomics-sections starts from PCA-plot of the results, and the LC-MS/MS results are not mentioned/shown. This is a major problem for the manuscript. These results should be presented as tables/supplementary tables and the results opened up.

Response: Thank you for your suggestions. The LC-MS/MS results of proteins identified have been added to Supplementary Material Table S2 in the revised manuscript.

Q2: Further, the raw data should be deposited into a publicly available proteomics repository like PRIDE (ProteomeXchange).

Response: The mass spectrometry proteomics data have been deposited to the ProteomeXchange Consortium (<http://proteomecentral.proteomexchange.org>) via the

iProX partner repository with the dataset identifier PXD027485. The link for reviewer is <https://www.iprox.cn/page/PSV023.html?url=1627006536055LeZN>, and Password is SgYy.

Q3: Fig 2B is not very informative in its current form, and Fig 2C is not needed, it just shows the number of DEPs already mentioned in the results. It would be much more informative to provide details on the DEPs as a table.

Response: Thanks for your comments. We have re-organized the Figure 2 (Page 33, lines 10-11). And the detail information of DEPs has been presented in Supplementary Material Table 3 in the revised manuscript.

Q4: Fig 3-6 are somewhat difficult to follow and the text in Results related to these is partly repetitive and could be significantly shortened.

Response: We have re-organized Fig 3-6 based on the re-analysis results according to your suggestions (Q9). Some repetitive sentences have been deleted in the revised manuscript (Page 9-10).

Q5: Re protein identification and quantification results: how many proteins were identified from serum samples?

Response: A total of 879 proteins were identified from serum samples. This information has been added to the revised manuscript (Page 6, line 19-20).

Q6: How reproducible was the depletion method used?

Response: ProteoPrep® Blue Albumin & IgG Depletion Kit was used in this study to deplete albumin and IgG, which is commonly used commercial kits in many studies. (*J Clin Invest.* 2020. 2069.)

Q7: In Mat+Met why was precursor mass tolerance 20 ppm, was the MS not properly calibrated?

Response: Peptide identification increased and FDR remained low in a large precursor tolerance search. And peptide identification in small precursor tolerance searches is highly conserved in larger search windows. (*Nature Medicine.* 2020. 769; *Journal of Proteomics.* 2013. 375.). MaxQuant performed a second accurate search based on the data quality.

Q8: How was the data processed after MaxQuant search? Something is mentioned in 'Statistical analysis' but many details are missing.

Response: The protein expression matrix quantized by MaxQuant was subjected to normalize by normalizeBetweenArrays function of the R package limma, which is an optimal normalization method commonly used in proteomics data analysis. (*Nature*. 2014. 513) Normalized protein abundance was log2-transformed and used in all quantitative analysis. As your suggestion, this information has been added in the revised manuscript (Page 18, lines 13-16).

Q9: How e.g were missing values taken into account in the analysis? It is written that if protein was quantified in two samples it was taken into statistical analysis. Thinking on the number of samples in proteomics analysis (90 in total) having a valid value in 2/90 samples is really low posing questions on the quality of the results.

Response: We thank the reviewer for raising this important issue. We reanalyzed the data with a total of 396 proteins, which were identified with fewer than 40% missing values from serum samples. The Result section has also been updated (Figures 2-6; Page 21, line 5-6).

Q10: Why did the authors choose proteins for ELISA validation 'randomly'?

Response: The aim of the ELISA validation confirms the expression value of proteomic analysis. In consideration of the availability of samples and ELISA kits, we

randomly selected 4 proteins from 12 biomarker candidates to validate the reliability of proteomic analysis.

Q11: Is CRP really a random choice and a new finding related to sepsis?

Response: CRP is traditional biomarkers for sepsis, which has been widely used in clinical practice. The identification of CRP in the study was further confirmed the results of our proteomic analysis, validating the specificity and research value of other identified biomarker candidates for sepsis.

Q12: How many of the proteins found to be DEPs in this study have previously been linked to sepsis?

Response: The 12 proteins (TKT, SEPP1, FIBA, FIBB, NUCB1, S10A9, CRP, AACT, A2GL, SAA2, SAA1, LBP) identified in this study (Figure 6B-D) were all biomarker candidates for sepsis. The information about these biomarker candidates linked to sepsis has been discussed in Discussion Section of the revised manuscript (Page 14, lines 1-3, lines 11-14 and 17-22).

Q13: What is the main novelty of the current study?

Response: In this study, we use both proteome-wide differential analysis and protein co-expression network to gain insights into changes in individual proteins as well as networks of proteins in sepsis. Our results showed the major characteristics of sepsis, such as the inflammatory response and wound healing. The importance of plasma lipoprotein-related processes and lipid metabolism in sepsis was revealed in this study.

August 3, 2021

RE: Life Science Alliance Manuscript #LSA-2021-01091R

Prof. Yinghe Xu
Taizhou Central Hospital (Taizhou University Hospital)
999 Donghai Avenue
Taizhou 318000

Dear Dr. Xu,

Thank you for submitting your revised manuscript entitled "Serum proteomics reveals disorder of lipoprotein metabolism in sepsis". We would be happy to publish your paper in Life Science Alliance pending final revisions necessary to meet our formatting guidelines. Please also incorporate the remaining requests made by Reviewer 3.

- please add ORCID ID for secondary corresponding author-they should have received instructions on how to do so
- please add the Twitter handle of your host institute/organization as well as your own or one of the first author in our system
- please upload your supplementary figure as a single file as well
- There is a callout for Figure 6F in the manuscript text, although the actual Figure does not have this panel, please revise
- please add a callout for Figure 6D to your main manuscript text

LSA now encourages authors to provide a 30-60 second video where the study is briefly explained. We will use these videos on social media to promote the published paper and the presenting author. Corresponding or first-authors are welcome to submit the video. Please submit only one video per manuscript. The video can be emailed to contact@life-science-alliance.org

A. FINAL FILES:

B. MANUSCRIPT ORGANIZATION AND FORMATTING:

Sincerely,

Reviewer #1 (Comments to the Authors (Required)):

The authors have answered all of my comments.

Reviewer #3 (Comments to the Authors (Required)):

The authors have addressed most of my previous comments/concerns, but I still miss details in proteomics data analysis. Now they mention which software they used for data analysis after MaxQuant, but how e.g missing value imputation was done is not described. Further, is the number of identified proteins (879) based on LFQ values? LFQs are normalized intensities and they do not tell exactly the number of identifications, that is a different column in MaxQuant results; many times low abundant raw intensities turn into zero during normalization. Was there any cut off how many unique peptides/protein was required for reliable identification or was default setting used? I think it would be important to add this type of details to mat+met.

To check the quality of the data I uploaded MaxQuant data into Perseus and did analysis there, I got slightly different number of DEPs but that is probably due to different filtering settings used, but overall the data quality looked fine and the PCA showed clear group separation.

Dear Reviewers,

Thank you very much for spending your valuable time reviewing our manuscript. We have tried our best to address the issues raised by each of you and hope that we have addressed all your concerns regarding this work.

Sincerely,

Dr. Yinghe Xu

Response to the comments raised by Reviewer #3

Reviewer #3

Comments

Q1: The authors have addressed most of my previous comments/concerns, but I still miss details in proteomics data analysis. Now they mention which software they used for data analysis after MaxQuant, but how e.g missing value imputation was done is not described.

Response: As you suggested, the information of missing value imputation has been added in the revised Materials and Methods section (Pages 18, lines 18-19).

Q2: Further, is the number of identified proteins (879) based on LFQ values? LFQs are normalized intensities and they do not tell exactly the number of identifications, that is a different column in MaxQuant results; many times low abundant raw intensities turn into zero during normalization.

Response: Sorry for the confused expression. A total of 879 proteins were identified from serum samples. We have corrected it in revised manuscript (Pages 6, line 19).

Q3: Was there any cut off how many unique peptides/protein was required for reliable identification or was default setting used? I think it would be important to add this type of details to mat+met.

Response: Thank you for your comments. A minimum peptide ratio count of two were required, and only unmodified peptides were used for relative quantification. This information has been added to the revised manuscript (Page 18, lines 12-13).

To check the quality of the data I uploaded MaxQuant data into Perseus and did analysis there, I got slightly different number of DEPs but that is probably due to different filtering settings used, but overall the data quality looked fine and the PCA showed clear group separation.

August 10, 2021

RE: Life Science Alliance Manuscript #LSA-2021-01091RR

Prof. Yinghe Xu
Taizhou Central Hospital (Taizhou University Hospital)
999 Donghai Avenue
Taizhou 318000
China

Dear Dr. Xu,

Thank you for submitting your Research Article entitled "Serum proteomics reveals disorder of lipoprotein metabolism in sepsis". It is a pleasure to let you know that your manuscript is now accepted for publication in Life Science Alliance. Congratulations on this interesting work.

DISTRIBUTION OF MATERIALS:

Again, congratulations on a very nice paper. I hope you found the review process to be constructive and are pleased with how the manuscript was handled editorially. We look forward to future exciting submissions from your lab.

Sincerely,
